# Impact of Cations (Na^+^, K^+^, Mg^+2^) and Anions (F^−^, Cl^−^, SO_4_^2−^) Leaching from Filters Packed with Natural Zeolite and Ferric Nanoparticles for Wastewater Treatment

**DOI:** 10.3390/ijerph18168525

**Published:** 2021-08-12

**Authors:** Evelyn Maria Miramontes-Gutierrez, Jesus Manuel Ochoa-Rivero, Hector Osbaldo Rubio-Arias, Lourdes Ballinas-Casarrubias, Beatriz Adriana Rocha-Gutiérrez

**Affiliations:** 1College of Animal Science and Ecology, Autonomous University of Chihuahua, Chihuahua 31453, Mexico; evymmg96@gmail.com; 2Instituto Nacional de Investigaciones Forestales, Agricolas y Pecuarias (INIFAP), Campo Experimental La Campana, Aldama, Chihuahua 32910, Mexico; 3College of Animal Production and Ecology, Autonomous University of Chihuahua, Chihuahua 31453, Mexico; hrubioa@uach.mx; 4College of Chemistry, Autonomous University of Chihuahua, Circuito Universitario S/N, Chihuahua 31125, Mexico; mballinas@uach.mx

**Keywords:** natural minerals, Mexican zeolites, leakage, positive ions, negative ions

## Abstract

Natural zeolites have been employed to adsorb contaminants in water. This study is aimed to evaluate the cation and anion leaching from the zeolite after the wastewater was passed through filters packed with a natural zeolite (heulandite-CaAl_2_Si_7_O_18_·6H_2_O). Eight treatments were evaluated in a 2 × 2 × 2 factorial treatment design. Factor A was the zeolite with two levels: 127 g and 80.4 g. Factor B was the nanoparticles with two levels: one bag (3.19 g) and two bags (6.39 g); and Factor C was the use of a magnet: with and without. There were two replications; hence, a total of 16 filters were employed. The water was obtained from a municipal wastewater treatment plant (MWTP). The cations (Na^+^, K^+^; Mg^+2^ and Ca^+2^) and anions (F^−^, Cl^−^ and SO_4_^2−^) were measured before (influent = IW) and after filtering (effluent = EW) three times. All treatments leached the cations Na^+^ (EW in a range of 175 to 232 ppm), K^+^ (EW in a range of 15.4 to 33.2 ppm), and Mg^+2^ (EW in a range of 7.40 to 10.8 ppm) but did not leach Ca^+2^. Likewise, the treatments leached the anions F^−^ (EW in a range of 7.59 to 8.87 ppm), Cl^−^ (EW in a range of 85.9 to 120 ppm), and SO_4_^2−^ (EW in a range of 139 to 146 ppm). We conclude that this natural zeolite leaches cations (except Ca^+2^) and anions in MWTP passed through filters. Therefore, its application in wastewater treatment should be considered for purposes such as agriculture and animal production and not for drinking water.

## 1. Introduction

Treated wastewater can be reused for specific purposes like garden irrigation, toilet water, industrial processes, and others [1], which could significantly decrease freshwater use [2]. Some adsorption methodologies are efficient and economical in removing contaminants from treated wastewater and potable water. For instance, natural zeolite has been used in packed filters to remove certain metals and metalloids from surface water [3,4,5,6], to diminish cations from water for industrial use [7] and potable water [8], to absorb ammonium in soils to reduce water contamination [9], and to reduce salinity [10], among other applications. The first use of natural zeolite as an inorganic ion exchanger was documented 50 years ago [11], followed by the study of Mercer et al. [12], who worked with clinoptilolite to adsorb NH_4_^+^ ions from water.

The importance of reusing treated water is magnified considering the scarcity of freshwater and the contamination of water reservoirs, the latter having increased demand for freshwater in other countries. The problem of water shortage is magnified among populations living in arid and semiarid environments, where water supply is always challenging [13]. Moreover, there is strong competition among industry, agriculture, animal production, urban centers, and others to access freshwater [14]. Unfortunately, the inhabitants of many environments do not consider freshwater as a finite and prized resource, considering it as just another commodity. A group of global experts has identified 19 solutions to the water crisis, among which the use of treated wastewater was considered an essential factor in reducing freshwater use and consequently advancing self-sufficiency [15].

Northern Mexico is an arid and semiarid region where inhabitants have problems accessing freshwater. For instance, the city of Chihuahua in the State of Chihuahua, with approximately one million inhabitants, depends entirely on groundwater as a water source. The lack of freshwater is aggravated by the presence of atypical droughts, which are common to these environments. Owing to this situation, the municipality has built two plants to collect wastewater. Treated wastewater can contribute to reducing the city’s freshwater requirements.

Several studies are showing the ion exchange capacity of the zeolite applied for water treatment. However, there is a lack of information regarding the ions leaching from this material and the interaction with nanoparticles into the water. To our knowledge, there is no evidence to date that natural zeolite leaks ions into the water when it is used as a decontamination filter. In addition to using natural zeolites, the technology of nanoparticles had been studied extensively to document the adsorption of metals in water by applying a magnetic field [16,17]. The aim of this research was to determine the quantities of cations: Na^+^, K^+^, Mg^+2^, Ca^+2^, and anions: F^−^, Cl^−^ SO_4_^2−^, and NO_3_^−^ leached from natural zeolite in filters added with ferric nanoparticles after passing wastewater. These ions were selected due to their reported importance in water-reuse quality for agricultural purposes. Our research hypothesis is that certain zeolites leak ions into water being filtered. Thus, this material requires treatment before being used to decontaminate water.

## 2. Materials and Methods

The study was carried out in the laboratory at La Campana Research Center of the National Research Institute for Forestry, Agriculture, and Animal Production (INIFAP-Mexico) located in the city of Aldama, Chihuahua, Mexico. The natural zeolite was supplied by a commercial mine located in the Municipality of Aldama, Chihuahua, situated in the next polygon; latitude 28°58′12′′ N; longitude 105°54′00′′ W. Natural zeolite used in this study are generally characterized as a volcanogenic sedimentary mineral, and the mine that supplied the zeolite has the capacity to supply 100 tons daily. Prior to packing the filters, a zeolite sample was characterized chemically at the Mexican Geologic Service by X-ray diffraction (XRD) with mineralogical decomposition. Porosity was determined by X-ray computed tomography, and petrographic and permeability analyses were performed at the SLE laboratory in Center for Scientific Research and Higher Education at Ensenada (CICESE, by acronym in Spanish), Baja California, Mexico.

After collecting zeolite in the field, it was crushed and filtered through several sieves until a granulometry of 30 mm (no. 2 sieve) was obtained. The nanoparticles used were magnetic ferrites (Mn_x_Fe_3_^−^_x_O_4_) produced by the chemical co-precipitation method [18]. The nanoparticles were bagged in 3.19 g containers, and a sample was characterized at the Center for Research in Advanced Materials (CIMAV-Mexico) in Chihuahua City.

The water samples for filtering were collected at the Municipal Wastewater Treatment Plant (MWTP) north of Chihuahua City. This MWTP collects wastewater from domestic and commercial sources. The parameters potential hydrogen (pH), total dissolved solids (TDS), and electrical conductivity (EC) were quantified in situ using a HANNA potentiometer and in accordance with legally established Mexican standards (NOM-001-SEMARNAT-1996, NOM-127-SSA1-1994, and NOM-230-SSA1-2002). Na^+^, K^+^, Mg^+2^, and Ca^+2^ cations in the water were measured according to the American Society for Testing Materials D6919. Similarly, F^−^, Cl^−^, and SO_4_^2−^ anions were quantified following the US EPA 300 A. Cation and anion concentrations were determined by ion-exchange chromatography (DIONEX ICS-1100; Thermo Fisher Scientific, Waltham, MA, USA) at the Faculty of Chemistry of the Autonomous University of Chihuahua, Mexico. Before applying the samples to chromatography, they were decanted to eliminate microalgae.

Eight filters were assembled using crystalline tubes 5.1 cm wide and 10 cm long to receive 127 g of zeolite, and eight more filters were assembled with crystalline tubes 5 cm wide and 7 cm long to receive 80.4 g of zeolite. The mineral utilized for this experiment was eolite (61.4%), a mixture of minerals that includes albite, orthoclase, muscovite, and calcite. The nanoparticle bags were placed in the upper part of the filters according to the treatments, and the filters were sealed with commercial silicon. The magnets were placed at the lower part outside of the filters. The permanent magnets were purchased in a retail store and were made of neodymium (Nd_2_Fe_14_B), with dimensions of 40 × 30 × 4 mm. The purpose of including magnets was to retain the nanoparticles to avoid leaching them. Thus, the experimental unit was a filter with different levels of the three factors under investigation. A total of eight treatments were evaluated as a result of a 2 × 2 × 2 factorial treatment design. Factor A was the quantity of natural zeolite with two levels: 127 g and 80.4 g. Factor B was the quantity of nanoparticles with two levels: one bag and two bags. Factor C was the use of a magnet with two levels: with and without the magnet. There were two replicates of each factor, so a total of 16 filters were constructed and filled for each treatment, which are detailed in Table 1.

A total of 500 mL of water, hereafter referred to as influent water (IW), was passed down through the filters three times. In other words, the same IW was passed through the filters three times (no-recirculation) by separating them. After each filtration, the water was collected, hereafter referred to as effluent water (EW). The quantified parameters in the IW were compared to those in the EW to identify differences in concentrations.

The statistical analysis consisted of an independent ANOVA for each filtration in the context of a 2 × 2 × 2 factorial design. All the analyses employed a confidence level of 95% so that α = 0.05.

## 3. Results

The chemical characterization of the zeolite used in this study showed that it was composed at 61.4% of CaAl_2_Si_7_O_18_·6H_2_O, with minor amounts albite, orthoclase, quartz, cristobalite, muscovite, calcite, magnetite, and hematite (Table 2). This indicates that the zeolite used in this study was a heulandite type closely related to clinoptilolite-type zeolite. Porosity was 0.292582%, and permeability 8.4263 × 10^−6^.

### 3.1. The Parameters pH, EC and TDS

The IW used in this study had a pH of 7.5. Figure 1a shows that pH levels ranged from 7.5 in T8 in the second filtration to 7.8 in T1 in the second filtration. The IW used in this study had an electrical conductivity (EC) of 1100 mS cm^−^^1^. Figure 1b shows that EC ranged from 980 mS cm^−^^1^ in T2 in the third filtration to 1370 mS cm^−^^1^ in T8 in the first filtration. The EC level was lower after the second and third filtrations than the level after the first treatment. Total dissolved solids (TDS) in IW measured 850 ppm. The behavior of this parameter was similar to that of EC. In fact, TDS and EC strongly correlate. Figure 1c shows that the TDS level in the EW after the first filtration was higher than the initial concentration in all treatments. Following the second and third filtrations, the TDS levels were generally the same as those in the IW.

### 3.2. Leaching Na^+^, K^+^, Mg^+2^, and Ca^+2^ Cations

Na^+^ levels in the IW were 148 ppm on average, meaning that this water could be used for irrigation. Figure 2a shows that after the first filtration, Na^+^ ranged from 209 ppm in treatment 4 to 232 ppm in treatment 6. After the second filtration, it ranged from 195 ppm in treatment 4 to 202 ppm in treatment 7. The same trend was noted after the third filtration, with Na^+^ ranging from 175 ppm in treatment 2 to 199 ppm in treatment 3. Figure 2a shows Na^+^ increased after all the treatments. It was also evident that Na^+^ ion concentrations were slightly lower in EW after the second and third filtrations. K^+^ content in the IW was on average 16.0 ppm. After the first filtration, K^+^ content ranged from 27.5 ppm in treatment 2 to 33.2 ppm in treatment 6, indicating that K^+^ was leached in all treatments. After the second filtration, K^+^ content ranged from 15.9 ppm in treatment 6 to 24.5 ppm in treatment 5. After the third filtration, K^+^ content ranged from 15.4 ppm in treatment 7 to 26.8 ppm in treatment 5. Figure 2b shows clearly that in most treatments, K^+^ increased after filtration. Figure 2b also shows that the K^+^ ion concentration after the second and third filtrations is slightly lower than the concentrations after the first filtration.

The Mg^+2^ levels in the IW was on average 6.99 ppm. Figure 2c shows that after the first filtration, the Mg^+2^ ranged from 7.40 ppm in treatment 4 to 8.63 ppm in treatment 1. After the second filtration, the concentration ranged from 7.82 ppm in treatment 2 to 9.33 ppm in treatment 3, and after the third filtration, Mg^+2^ ranged from 8.99 ppm in treatment 2 to 10.8 ppm in treatment 7. Figure 2c shows that Mg^+2^ clearly increased with all treatments. It should also be noted that Mg^+2^ concentrations continued to increase after the second and third filtrations. The quantity of Mg^+2^ leached was small compared to that of other elements. The Ca^+2^ level in the IW was 53.5 ppm. After the first filtration, Ca^+2^ ranged from 35.5 ppm in treatment 6 to 45.6 ppm in treatment 8. Figure 2d shows that Ca^+2^ was adsorbed in all treatments during the first filtration. After the second filtration, the concentration of Ca^+2^ ranged from 40.4 ppm in treatment 1 to 51.0 ppm in treatment 8. The treatments continued to adsorb Ca^+2^ instead of leaching it. After the third filtration, the concentration of Ca^+2^ ranged from 46.7 ppm in treatment 1 to 54.3 ppm in treatment 3. Figure 2d shows that after the first pass of IW for the filters, there was no leaching effect on Ca^+2^ after the first filtration and that the concentration of Ca^+2^ increased with the second and third filtrations.

### 3.3. Leaching F^−^, Cl^−^, and SO_4_^2−^ Anions

The ANOVA did not detect statistically significant differences for Factor A (*p* > 0.05), Factor B (*p* > 0.05), or the interaction between the two (*p* > 0.05) for any of the anions in any of the filtrations. F^−^ levels in the IW in this study were on average 1.84 ppm. Figure 3a shows that after the first filtration, the concentration of F^−^ was higher than the original level of 1.84 ppm, ranging from 7.59 ppm in treatment 3 to 8.87 ppm in treatment 7. After the second filtration, the concentration of F^−^ ranged from 6.84 ppm in treatment 1 to 8.55 ppm in treatment 4. After the third filtration, the concentration of F^−^ ranged from 6.88 ppm in treatment 1 to 8.84 ppm in treatment 2. The Cl^−^ level in the IW was on average 80.8 ppm. Figure 3b shows that the Cl^−^ level after the first filtration was higher than the original level, ranging from 101 ppm in treatment 4 to 120 ppm in treatment 8. After the second filtration, the concentration of Cl^−^ ranged from 89.0 ppm in treatment 2 to 98.9 ppm in treatment 7. After the third filtration, the concentration of Cl^−^ ranged from 85.9 ppm in treatment 2 to 100 ppm in treatment 5. The SO_4_^2−^ level in the IW was on average 133 ppm. Figure 3c shows that after the first filtration, SO_4_^2−^ was slightly above the original level, ranging from 139 ppm in treatment 2 to 144 ppm in treatment 7. The concentration after the second filtration ranged from 139 ppm in treatment 4 to 146 ppm in treatment 7. After the third filtration, SO_4_^2−^ ranged from 139 ppm in treatment 1 to 145 ppm in treatment 7.

## 4. Discussion

Mexico has large zeolite reserves due to its geology [19], and different types of zeolite have been used for diverse purposes, including adsorption of heavy metals from water [20] and even in human health [21]. Natural zeolites have a tetrahedral framework (SiAl) with four adjacent oxygen atoms (SiAl)O_4_ with negative charges, where the substitution of Si^4+^ for Al^3+^ allows exchangeable cations, such as Na^+^, K^+^, Ca^+2^, and Mg^+2^. That singularity of ion exchange is known as stoichiometric replacement, in which one ion in the solid phase is substituted by another in the liquid phase. Therefore, the reactions occur between cations (Na^+^, K^+^, Ca^+2^, Mg^+2^) in the zeolite and between the cations in the water. In other words, adsorption/desorption are chemical phenomena that occur on the particle’s surface and the micro environmental interface. Some researchers define this process as amphoteric [22] because it is possible to adsorb cations at specific and varying pH levels. 

### 4.1. The Parameters pH, EC, and TDS

It is well documented that the pH level plays an important role in the adsorptive capacity of natural zeolite [23]. Most research on the adsorptive capacity of zeolite has found that zeolite is at maximum efficiency at low pH levels (i.e., below 5.0). Nevertheless, other studies, such as that of Ji et al. [24], found that the highest NH4 removal rates were at pH 6.0, while Hong et al. [25] found that the highest removal rates for zeolite are under alkaline conditions. Onyango et al. [26] theorized that pH is the main factor affecting adsorption. In another study, Assefa and Adugna [27] found that the maximum adsorption rates for Ca^+2^ and Mg^+2^ were at pH 6.5. It is understandable that the EC of treated wastewater is higher than that of than freshwater because of the presence of dissolved salts. The main objective of treating wastewater is to obtain acceptable salinity levels so that wastewater can be reused for irrigation and other purposes. The EC levels of the IW used in this study met Mexican norms that establish 2000 mS cm^−^^1^ as adequate for irrigation (NOM CCA/032-ECOL/1993). It is clear that the EC level increased after the first filtration, meaning that there was a salt-leaching effect. Wibowo et al. [10] analyzed the capacity of a natural zeolite collected in Sukabumi, West Java, Indonesia, to reduce salinity and concluded that this capacity correlated to the amount of zeolite used. For TDS, Palomo et al. [28] found TDS levels of 893 ppm in a study of residual water in Ciudad Juarez, Chihuahua, Mexico, while a study of the Luis L. Leon Dam in the State of Chihuahua, Mexico, reported TDS levels of 863 ppm [29].

### 4.2. Leaching of Cations Na^+^, K^+^, Mg^+2^, and Ca^+2^ from Zeolite

It is well documented that the Na^+^ ion can form sodium salts that are highly soluble in water, making high concentrations of these salts harmful to human health and certain environments. Our results clearly show that after the first filtration, all the treatments leached this cation, and the amount in the EW exceeded in some cases 200 ppm, a level that is too potentially harmful to be released into streams and other waterways or used for irrigation. Although there are no Mexican standards for acceptable amounts of sodium in irrigation water, it is clear that water with high amounts of Na^+^ can increase soil impermeability and decrease hydraulic conductivity, affecting soil capacity to hold rainwater. In fact, Na^+^, Ca^+2^, and Mg^+2^ are the most important cations in calculating the equation known as the sodium adsorption ratio (SAR), which is used to characterize the sodium content of irrigation water [30]. The SAR ratio results from dividing the Na^+^ concentration by the square root of half of the Ca^+2^ and Mg^+2^ concentrations. The following Equation (1) describes how to calculate this ratio:(1)$$\mathrm{SAR}=\frac{{\mathrm{Na}}^{+}}{\sqrt{\frac{{\mathrm{Ca}}^{+2}{+\mathrm{Mg}}^{+2}}{2}}}$$

The SAR value for the original treated wastewater used in this study was 27.0, which the U.S. Department of Agriculture would classify as C3S4 type water. This level of sodium represents a hazard to irrigation water. According to the present study results, all treatments increased the SAR value, thus affecting the water quality of the treated wastewater. Wang et al. [31] used natural zeolite to remove Na^+^ from coal seam gas (CSG) water samples and found that natural zeolite removed only a slight quantity of Na^+^. However, they reported an increase in Ca^+2^ and Mg^+2^. They concluded that natural zeolite does have not the capacity to remove Na^+^, but its ion exchange capacity with Ca^+2^ and Mg^+2^ decreased the SAR value and consequently improved water quality.

Our study showed that K^+^ was not adsorbed in the different treatments with zeolite, nanoparticles, and a magnet and instead, K^+^ was leached. Our findings disagree with those of Jaskūnas et al. [32], who stated that K^+^ adsorption is followed by desorption of other cations, such as Ca^+2^, Na^+^, and Mg^+2^. Nevertheless, the finding that K^+^ ions are leached from natural zeolite is important in relation to the usefulness of this type of zeolite as an inorganic fertilizer given that plants take up K^+^ from the soil [33]. After nitrogen and phosphate, K^+^ is the most important fertilizer.

This was confirmed by Jaskūnas et al. [32], who reported that Mg^+2^ ions were desorbed in small quantities. Along with Ca^+2^, Mg^+2^ is responsible for water hardness problems that have been reported in other countries. However, human consumption of hard water has been related to lower cardiovascular disease [34]. It is clear that zeolite could be used as a low-cost adsorbent. However, the natural zeolite used in this study has the property of leaching, which must be considered when using it to decontaminate water.

As noted above, Ca^+2^ and Mg^+2^ are responsible hardening water, and researchers like Cinar and Beler-Baykal [35] have reported promising methods to soften water by removing 11 mg of Ca^+2^ per gram water using clinoptilolite. Our results are similar to those reported by Mabovu [36], who worked with an untreated Turkish clinoptilolite and found that Ca^+2^ leached about 162% and Mg^+2^ about 37% from the zeolite’s adsorbent structure. The author hypothesized that univalent cations like K^+^ are first adsorbed, while divalent cations like Ca^+2^ and Mg^+2^ are leached in the process.

### 4.3. Leaching of Anions F^−^, Cl^−^, and SO_4_^2−^

F^−^ is considered beneficial to humans at a level of 0.7 ppm but hazardous beyond 1.5 ppm [37]. In addition to ingesting it through water, humans consume F^−^ in food and are exposed to it in workplaces and other contexts [37].

It is generally known that natural zeolite has a capacity to adsorb anions [38,39], which was confirmed by Pérez-Escobedo et al. [40] and Habuda-Stanić et al. [41]. However, our results clearly show that the natural zeolite used in this study has the capacity to leach F^−^ instead of adsorbing it. Our results are confirmed by the research performed by Rahmani et al. [42], who reported that natural zeolite does not show any adsorption power in water with a concentration of F^−^ of 3.0 ppm, 2.8 ppm, and 2.6 ppm after comparing the results with two modified zeolites. In contrast, Sun et al. [43] demonstrated that it was possible to reduce F^−^ from a concentration of 10 mg L^−1^ to 1.0 mg L^−1^ using natural stilbite zeolite. Similar results were reported by Gómez-Hortigüela et al. [44], who tested Ethiopian mordenite-type zeolite. Their results showed a high level of defluoridation in F^−^-rich waters. Most researchers have recommended modifying natural zeolite; for instance, Zhang et al. [45] suggested a chemical modification of natural zeolite to enhance fluoride removal in aqueous solutions.

Natural zeolite has a low capacity to remove or leach Cl^−^ ions from water. However, Osio-Norgaard and Srubar [46] tested three zeolites (faujasite, chabazite, and mordenite) and demonstrated that faujasite and chabazite clearly have the capacity to adsorb Cl^−^. Our results concerning Cl^−^ agree with those of Mabovu [36], who found that Cl^−^ leached from a brine solution, which they attributed to the high pH level. There has been little research into SO_4_^2−^ removal or leaching from natural zeolites in aqueous solutions. Nevertheless, more information is available concerning modified zeolite. For example, Vidal et al. [47] tested three different modified zeolites with surfactants (SMZ-ODA, SMZ-Oleyamine, and SMZ-HDTMA) to remove sulfate and found that 80% was removed after six hours of contact without statistical differences among the three modified zeolites.

## 5. Conclusions

The natural Mexican zeolite used in this study leached the cations Na^+^, K^+^, and Mg^+2^ but not Ca^+2^. Similarly, the anions F^−^, Cl^−^, and SO_4_^2−^ were leached after treated water was filtered through zeolite without showing any sign of adsorption. It was clear that the amount of zeolite used in this research and the use of nanoparticles and magnets did not affect the quantity of cations and anions leached into treated water. Therefore, we highly recommend investigating this phenomenon with other natural zeolites to optimize their use in different processes. If leaching occurs with all-natural Mexican zeolites, modifying the methods to achieve higher cation and anion adsorption capacity will be necessary. It was clear that effluent water coming from the filters contained a higher amount of Na^+^, K^+^, Mg^+2^, and F^−^ than the concentration in influent water, although none of these elements were contained in the natural zeolite. Hence, it is highly recommended to identify the origin of the presence of ions detected in water. The water quality did not improve after passing through the filters packed with natural zeolite. Hence, the treated water may be used only for agriculture purposes where calcium needs to be removed for water.

## Figures and Tables

**Figure 1 ijerph-18-08525-f001:**
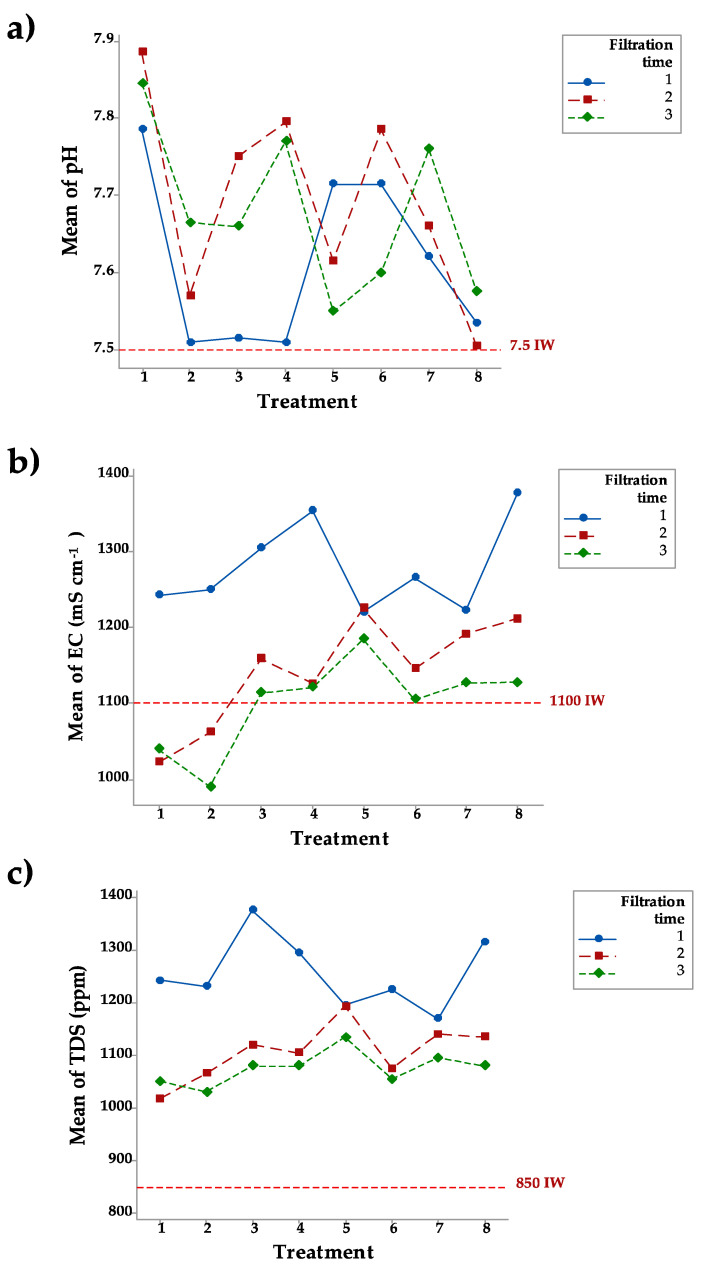
Levels of pH (**a**), EC (**b**), and TDS (**c**) in the eight treatments and three filtrations.

**Figure 2 ijerph-18-08525-f002:**
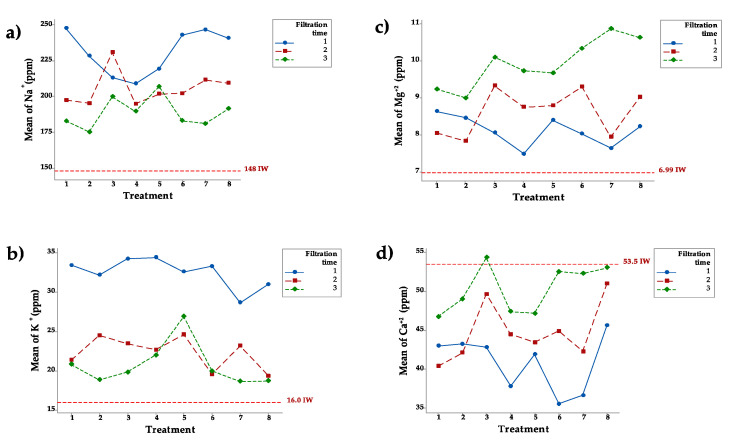
Concentration of Na^+^ (**a**), K^+^ (**b**), Mg^+2^ (**c**), and Ca^+2^ (**d**) in eight treatments after three filtrations.

**Figure 3 ijerph-18-08525-f003:**
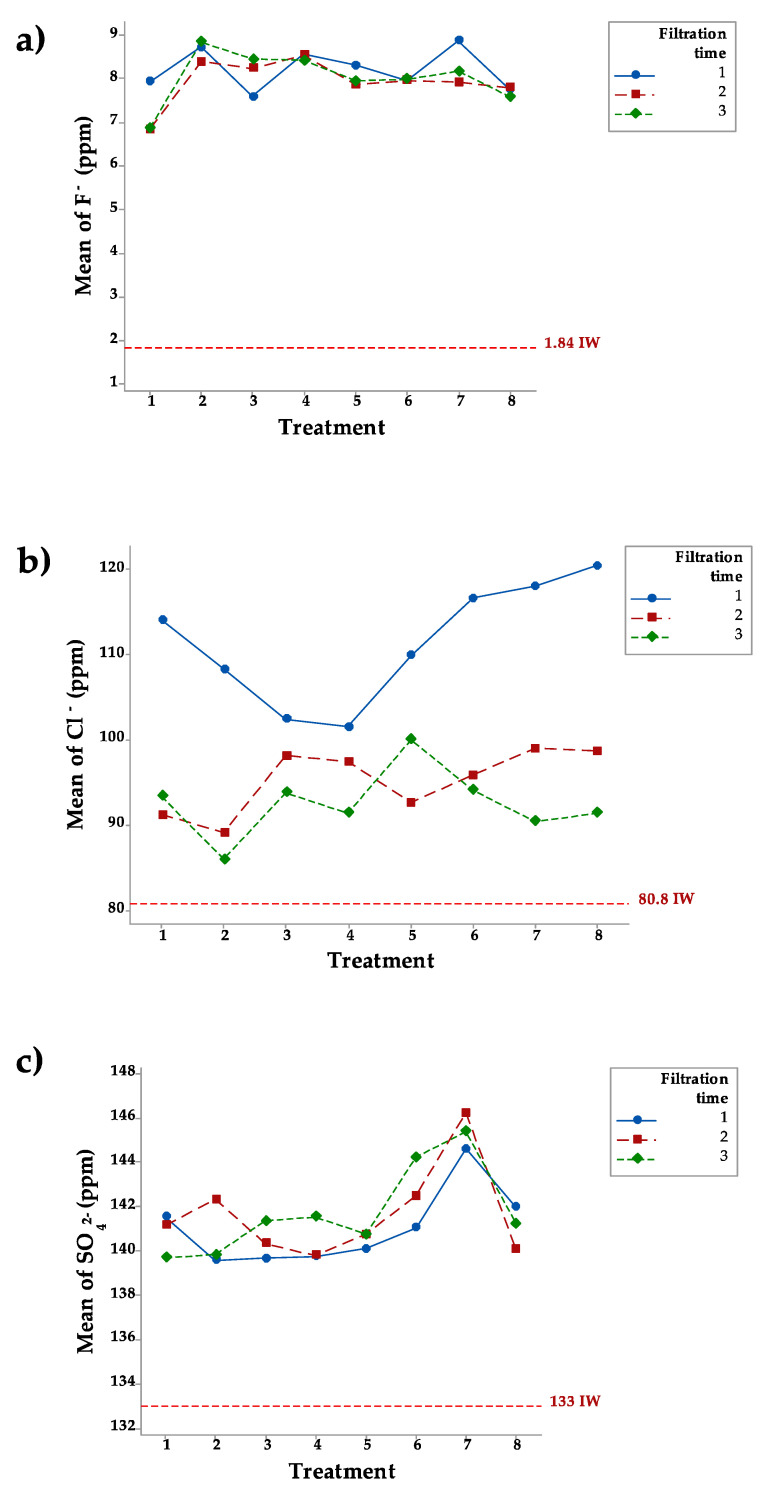
Concentrations of F^−^ (**a**), Cl^−^ (**b**), and SO_4_^2−^ (**c**) after three filtrations in eight treatments.

**Table 1 ijerph-18-08525-t001:** Treatments evaluated with a 2 × 2 × 2 factorial design.

Number	Treatment
T1	127 g Zeolite + 1 nanoparticle bag + with magnet
T2	127 g Zeolite + 1 nanoparticle bag + without magnet
T3	127 g Zeolite + 2 nanoparticle bags + with magnet
T4	127 g Zeolite + 2 nanoparticle bags + without magnet
T5	80.4 g Zeolite + 1 nanoparticle bag + with magnet
T6	80.4 g Zeolite + 1 nanoparticle bag + without magnet
T7	80.4 g Zeolite + 2 nanoparticle bags + with magnet
T8	80.4 g Zeolite + 2 nanoparticle bags + without magnet

**Table 2 ijerph-18-08525-t002:** Chemical characterization of the natural zeolite used in the study.

Mineral Species	Chemical Formula	Percentage
Heulandite	CaAl_2_Si_7_O_18_·6H_2_0	61.4
Albite	NaAlSi_3_O_8_	10.1
Orthoclase	KAlSi_3_O_8_	9.78
Quartz	A–SiO_2_	6.77
Cristobalite	SiO_2_	6.55
Muscovite	KAl_2_Si_3_O_10_(OH)_2_	3.90
Calcite	CaCO_3_	1.40
Magnetite	Fe_3_O_4_	Traces
Hematite	Fe_2_O_3_	Traces

## Data Availability

Data are available upon request, but if used in another manuscript, the authors wish to be included as co-authors.

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
