# Peer review of "Impact of Cations (Na+, K+, Mg+2) and Anions (F, Cl, SO42−) Leaching from Filters Packed with Natural Zeolite and Ferric Nanoparticles for Wastewater Treatment"

_ijerph, 2021, doi:10.3390/ijerph18168525_

Round 1

Reviewer 1 Report

The manuscript "Leaching cations ..." has potential but needs some revision before it is ready for publication.  It is well written but it is very confusing and lacks a clear objective.  Some specific comments and questions follow:

Line 2: Change "in" to into. Delete Ca because it was not leached but was adsorbed. Or rewrite the title to make it clear what happened.

Line 38: decontaminate? In what way did the proposed treatment decontaminate the water? All it has done is add some ions to the water. 

Line 39: Can you list a single "other purpose" ?

Line 76:  "leached from wastewater..." All your data and graphs indicate that everything except Ca ions  leached from the zeolite filters into the wastewater not from the wastewater.  This sentence greatly adds to confusion. 

Line 83: The source of the "natural zeolite" needs some description.  What kind of rock was it?  Judging from the mineralogy (Table 2) it was probably some kind of vesicular volcanic rock probably rhyolite but it should be stated clearly. Is it a large deposit that could supply much more material or a rare occurrence?

Line 120 Describe the magnets.  Are they powerful electromagnets? And what were they supposed to accomplish?

Line 127: "After each filtration the water was collected ..." Was the same water that was collected passed through the filters two more times or was more influent water used each time? You must make this clear; very important.

Line 157  All the charts were impossible to read without the use of a strong magnifier.  I used a strong hand lens and still had trouble reading the numbers.  They all need to be enlarged.

Lines 167 through 220:  All the charts indicate that each of the zeolite treatments with or without the ferric nanoparticles and/or magnet leached each of the measured ions from the filter into the influent water (IW) except Ca ions. Why would anyone want to do that? The Na, K, Mg, F, and Cl levels are made much worse.  The Ca levels have improved and the effluent water (EW) is softer;  but the F levels are much worse and have become dangerous.

               You need to explain why almost all the leaching out of the zeolites occurs on the first treatment then it looks like some of the same ions are absorbed back into the zeolite filter on the second and third pass through the filter.  Or is additional untreated IW being passed through each time.  I can't tell from your "Methods" description.  In either case you need to explain why the numbers change so dramatically after each pass through the filters.

               You need to make it clear that most ions are not leaching out of the zeolite (Heulandite) but out of the other minerals in the filter.  For example Heulandite does not contain any Na or K so the Na must be leaching out of the albite and the K from the orthoclase and muscovite. I don't know where the Mg and anions are coming from.

Line 328 – 336. Interesting conclusions; but would you recommend using your tested zeolite mixture for any purpose?  What are the pros and cons?  

Author Response

Dear Reviewer,

The authors would like to express your gratitude to you for your excellent work. We have considered all suggestions and correcttions.

I attached the author's reply for your comments.

Warm regards.

Jesus Ochoa-Rivero

Reviewer 2 Report

The authors studied sorption/leaching ability of natural heulandite – itself, and in combination with ferrite nanoparticles – and its possible utilization in water decontamination. Although the motivation for this work is very topical and actual, the manuscript lacks novelty and brings only poor results reminding rather technical report. Therefore, I do not support the manuscript for publication in scientific journal.

Major objections are:

Why the authors used combination with magnetic nanoparticles? What was expected role of utilization of magnetic field? None of analysed ions are magnetic, and leaking of ferrite was not tested. Hypothesis/explanation of used experimental setup (including magnetic field) is missing.

Usually, metal ions were leaked from the material, and repeated treatment stabilized final concentration, which was mostly higher than in incoming water. However, significant leak was observed in the case of sodium and potassium, which are not present in formula of porous mineral (and probability of its leak from e.g. feldspar or muscovite is near to zero). What was source of Na/K? To study leaking capacity, treatment of raw material with pure (deionized) water or defined acid solution should be done instead of applying real water sample.

p4,l156 – TDS level is much higher after even third filtration (>1000) than in incoming water (850).

Although studied zeolite is presented as material for possible decontamination, no experiments with e.g. toxic transition metal ions were performed. Zeolite can potentially work as stable ion exchanger leaking e.g. H/Na/K/Mg/Ca and chemisorbing transition metal ions. Larger volumes of water compared to weight of zeolite bed should be treated to study potential use as ion exchanger/chemisorber.

Formal:

Values of concentrations are presented with unjustified accuracy – up to 5 valid digits. More analysis should be done than only “doublets”, and ESD should be given and represented e.g. by error-bars in charts. Starting level (concentration) should be given in charts, too.

Author Response

Dear Reviewer,

The authors would like to express your gratitude to you for your excellent work. We have considered all suggestions and corrections.

I attached the author's reply for your comments.

Warm regards.

Jesus Ochoa-Rivero

Reviewer 3 Report

Comments to Author:

Zeolites are used to adsorb contaminants from water. The investigators question this habit and study the potential leakage from cations and anions from natural zeolite filters combined with nanoparticles and the use of a magnet. Water was passed through the filters and effluent water was collected and analyzed for Na, K, Mg, Ca, and F, Cl and SO4 and pH, electrical conductivity, total dissolved solids. Indeed, they find leakage of cations and anions from the zeolite.

General comment

This is a very relevant study in the field of water treatment pointing out a contamination risk. The experiments are well performed. The chemical analyses are allright. The design and statistical analysis seem all right but may be presented with more clarity. English is generally OK.

Specific comments

Introduction

p2, line 45       remove ‘toilet water’ (once is enough)

p2, line 49       from ? potable water

p2, line 61       ‘has’, not shas

p2, line 77       ‘added’sounds strange, do you mean ‘mixed’ ? or ‘combined’ ?

p2, line 78       leak ‘ions’

p2, line 72/73 Please dedicate a sentence to the use of zeolites. The reader needs more   background here.

p2, line 72/73 Please dedicate a sentence to the use of nanoparticles and the magnet. The  reader needs more background here. A reference is not sufficient.

P2, line 76      Again, please dedicate a sentence to the choice of the  cations and anions.

p2, line 79       before ‘being’used

Materials and Methods

p3, line 96       Please give a reference here for the coprecipitation method.

p3, Table 1     Why did n’t you include the two levels of zeolite without the nanoparticles and the magnets in the design. They should be important controls.

p3, line 131     ‘through’ the filters, not for

p3, line 130/134 This alinea is kind of confusing. I think it is not a recirculation ? Please rephrase.

P3, line 130    All filters were tested with half a liter of  IW. What brings you to such a volume, what makes you choose three filtration rounds.

Results

You may present more details on the outcome of the statistics in relation to the design (perhaps in a table).  The focus of your description is very much on the differences between filtration rounds, but you may also include something on the difference, or lack of difference, between the actual treatment groups. The reader requires more grip on where the variation is going. For instance in Figure 3c (on the SO4) there is a difference between Treatment 7 and all other treatments, this may be described in the Results, and also discussed in the Discussion.

You report the initial parameter levels in the IW in the text. You may insert them in all figures. It makes comparison faster for the reader.

Discussion

You may pay more attention to the choice of and difference between treatment groups, not only the filtration rounds (See above)

Conclusions and Summary

The Summary presents more specifics on the statistics than the Results. The last sentence of the Summary is not really found in the Discussion and the Conclusions. Please harmonize.

Author Response

Dear Reviewer,

The authors would like to express their gratitute to the reviewer for your excellent work. We have considered all suggestions and corrections.

TITLE:

Leaching cations (Na+, K+; Mg+2, Ca+2) and anions (F-, Cl- SO42-) in treated wastewater using filters packed with natural zeolite and ferric nanoparticles

TITLE SUGGESTED:

Impact of cations (Na+, K+, Mg+2) and anions (F-, Cl-, SO42-) leaching from filters packed with natural zeolite and ferric nanoparticles for wastewater treatment

The authors would like to express their gratitude to the reviewers for their excellent work. We have considered all suggestions and corrections.

REVIEWER 3

Comments and Suggestions for Authors. Zeolites are used to adsorb contaminants from water. The investigators question this habit and study the potential leakage from cations and anions from natural zeolite filters combined with nanoparticles and the use of a magnet. Water was passed through the filters and effluent water was collected and analyzed for Na, K, Mg, Ca, and F, Cl and SO4 and pH, electrical conductivity, total dissolved solids. Indeed, they find leakage of cations and anions from the zeolite.

Authors' response.  The authors thank the observations of the reviewer.  

Comments and Suggestions for Authors. This is a very relevant study in the field of water treatment pointing out a contamination risk. The experiments are well performed. The chemical analyses are allright. The design and statistical analysis seem all right but may be presented with more clarity. English is generally OK.

Authors’ response. We appreciate your opinion about our work. We believe that the manuscript will be ready for publication after improving it with the excellent reviewer’s comments. We tried to attend the comments of reviewers in a proper way

Comments and Suggestions for Authors. p2, line 45 remove ‘toilet water’ (once is enough)

Authors’ response. We agree. It has been removed the words “toilet water”.

Comments and Suggestions for Authors. p2, line 49  from ? potable water

Authors’ response. We agree. It has been included the word “potable water”.

Comments and Suggestions for Authors. p2, line 61 ‘has’, not shas

Authors’ response. We OK with your comment. However, in this line the word is “has” not “shas”.

Comments and Suggestions for Authors. p2, line 77  ‘added’sounds strange, do you mean ‘mixed’ ? or ‘combined’ ?

Authors’ response. We agree.  It has been changed “added” for “combined” in the manuscript.

Comments and Suggestions for Authors. p2, line 78 leak ‘ions’

Authors’ response. We agree.  It has been included the “ions”. 

Comments and Suggestions for Authors. p2, line 72/73 Please dedicate a sentence to the use of zeolites. The reader needs more background here.

Comments and Suggestions for Authors. p2, line 72/73 Please dedicate a sentence to the use of nanoparticles and the magnet. The  reader needs more background here. A reference is not sufficient.

Comments and Suggestions for Authors. P2, line 76 Again, please dedicate a sentence to the choice of the cations and anions.

Authors’ response. The reviewer is correct. For this comments, we added this sentence in the manuscript “Several studies are showing the ion exchange capacity of the zeolite applied for water treatment. However, there is a lack of information regarding the ions leaching from this material and the interaction with nanoparticles into the water”.

Comments and Suggestions for Authors. p2, line 79 before ‘being’used

Authors’ response. We appreciate the reviewer´s observation. We have included “being” in the original manuscript.

Comments and Suggestions for Authors. p3, line 96 Please give a reference here for the coprecipitation method.

Authors’ response. We appreciate the reviewer´s comment. We have included the reference: Ochoa-Rivero, J.M.; Barrientos-Juarez, E. Elaboración de Nanopartículas. Desplegable para productores No. 56. CIRNOC-C.E. La Campana, Aldama, Chihuahua, México. 2018, pp. 1.

Comments and Suggestions for Authors. p3, Table 1 Why didn’t you include the two levels of zeolite without the nanoparticles and the magnets in the design. They should be important controls.

Authors’ response. The eight treatments came from a factorial treatment design 2 x 2 x 2, where the factor A was the zeolite with two levels; 80.4 and 127 g. The factor B was the nanoparticle with two levels; one bag and two bags. The reviewer is right. Instead of testing the two levels of nanoparticles (one bag and two bags) we should tested without nanoparticles and with nanoparticles (an amount). In future research, we will include these controls for this study we did not have them.

Comments and Suggestions for Authors. p3, line 131 ‘through’ the filters, not for

Authors’ response. We agree with your observation. I has been changed the word “for” by “through”.

Comments and Suggestions for Authors. p3, line 130/134 This alinea is kind of confusing. I think it is not a recirculation ? Please rephrase.

Authors’ response. We agree with your comment. In the manuscript, we improve this sentence and clarify that this experiment did not use recirculation.

Comments and Suggestions for Authors. P3, line 130    All filters were tested with half a liter of  IW. What brings you to such a volume, what makes you choose three filtration rounds.

Authors’ response. We appreciate reviewer’s comment. I choose this volume and filtration rates according to our last study with zeolite filters (Rubio-Arias et al., 2021).

Rubio-Arias, H.O.; Ochoa-Rivero, J.M.; Villalba, M.L.; Barrientos-Juarez, E.; De La Mora-Orozco, C.; Rocha-Gutierrez, B.A. Eliminating heavy metals from water with filters packed with natural zeolite of varying sizes. TyCA. 2021, 12, In press.

RESULTS

Comments and Suggestions for Authors. You may present more details on the outcome of the statistics in relation to the design (perhaps in a table).  The focus of your description is very much on the differences between filtration rounds, but you may also include something on the difference, or lack of difference, between the actual treatment groups. The reader requires more grip on where the variation is going. For instance in Figure 3c (on the SO4) there is a difference between Treatment 7 and all other treatments, this may be described in the Results, and also discussed in the Discussion.

Authors’ response. We think that the results among treatments is clear enough with the inclusion of the IW level in the charts. Therefore, we the authors did not believe that an additional table with outcome in treatments will be required.

Comments and Suggestions for Authors. You report the initial parameter levels in the IW in the text. You may insert them in all figures. It makes comparison faster for the reader.

Authors’ response. We agree. The initial parameter has been included in the graphs.

DISCUSSION

Comments and Suggestions for Authors. You may pay more attention to the choice of and difference between treatment groups, not only the filtration rounds (See above).

Authors’ response. We agree. The discussion has been improved.

CONCLUSIONS AND SUMMARY

Comments and Suggestions for Authors. The Summary presents more specifics on the statistics than the Results. The last sentence of the Summary is not really found in the Discussion and the Conclusions. Please harmonize.

We agree. The last sentence of the abstract had been changed. The other reviewer suggested something similar.

Warm regards.

Jesus Ochoa-Rivero

Round 2

Reviewer 1 Report

The revised manuscript is improved but could still be improved a bit more.  It seems to me that it should be made clear somewhere in the text, probably in the conclusions, that: 1. The filters used are not pure zeolite but a mixture of minerals that include albite, orthoclase, muscovite, and calcite.  The zeolite content is only 61.42 percent (Table 2). 2.  The filtered or treated water (EW) contains more Na, K, Mg, and F than the influent (IW) although none of these elements are contained within the zeolite (heulandite). and 3.  Water quality is not improved and the treated water would not be suitable for drinking.

Author Response

Dear Reviewer,

The authors would like to express your gratitude to the reviewer for your excellent work. We have considered all suggestions and corrections.

TITLE:

Leaching cations (Na+, K+; Mg+2, Ca+2) and anions (F-, Cl- SO42-) in treated wastewater using filters packed with natural zeolite and ferric nanoparticles

TITLE SUGGESTED:

Impact of cations (Na+, K+, Mg+2) and anions (F-, Cl-, SO42-) leaching from filters packed with natural zeolite and ferric nanoparticles for wastewater treatment

The authors would like to express their gratitude to the reviewers for their excellent work. We have considered all suggestions and corrections.

REVIEWER 1

Comments and Suggestions for Authors. The revised manuscript is improved but could still be improved a bit more. It seems to me that it should be made clear somewhere in the text, probably in the conclusions, that: 1. The filters used are not pure zeolite but a mixture of minerals that include albite, orthoclase, muscovite, and calcite. The zeolite content is only 61.42 percent (Table 2). 2. The filtered or treated water (EW) contains more Na, K, Mg, and F than the influent (IW) although none of these elements are contained within the zeolite (heulandite). and 3. Water quality is not improved and the treated water would not be suitable for drinking.

Authors' response. We agree with the reviewer. We added the following sentence in the revised version:– methods and materials “The mineral utilized for this experiment is besides zeolite (61.4 %) a mixture of minerals that include albite, orthoclase, muscovite, and calcite”. Also, we included the following sentence in the conclusion section “It was clear that effluent water coming from the filters contained more Na+, K+, Mg+2, and F- than the concentration in influent water. Although none of these elements were contained in the natural zeolite. Hence, it is highly recommended to identify where those elements are coming from”. For last comment, we included the following sentence in the conclusion “It is also concluded that water quality is not improved after passing across to the filters packed with natural zeolite. Hence, the treated water would not be suitable for drinking”.

Warm regards.

Jesus Ochoa-Rivero

Reviewer 2 Report

Compared to the previous version of the manuscript, the authors slightly improved the text and add some explanation, which better explains their motivation. However, they very probably made a mistake in units of magnetic field used in their experiments – it should be probably 0.5 T (tesla), and not 0.5 G (gauss), as 0.5 G corresponds to strength of the Earth’s magnetic field...

None of my objections was addressed – for accepting the manuscript, at least leaking experiments with pure (distilled) water should be done to show real “leaking” ability of the material, and not only sorption/ion exchange of ions from "real" water, as was studied.

To my final objection to the previous version – there was a misunderstanding between me and the authors. I recommended to decrease a number of valid digits, as 5 digits are too much and it is behind accuracy of analysis, as accuracy is usually 3 digits (not decimal places). I.e., use e.g. 276 ppm, 27.6 ppm, 2.76 ppm, and not 276.00 and 27.60...

For service to reader, starting level (concentration) of ions should be given in charts, too.

Author Response

Dear Reviewer,

The authors would like to express your gratitute to the reviewer for your excellent work. We have considered all suggestions and corrections.

TITLE:

Leaching cations (Na+, K+; Mg+2, Ca+2) and anions (F-, Cl- SO42-) in treated wastewater using filters packed with natural zeolite and ferric nanoparticles

TITLE SUGGESTED:

Impact of cations (Na+, K+, Mg+2) and anions (F-, Cl-, SO42-) leaching from filters packed with natural zeolite and ferric nanoparticles for wastewater treatment

The authors would like to express their gratitude to the reviewers for their excellent work. We have considered all suggestions and corrections.

REVIEWER 2

Comments and Suggestions for Authors. Compared to the previous version of the manuscript, the authors slightly improved the text and add some explanation, which better explains their motivation. However, they very probably made a mistake in units of magnetic field used in their experiments – it should be probably 0.5 T (tesla), and not 0.5 G (gauss), as 0.5 G corresponds to strength of the Earth’s magnetic field...

Authors' response.  We apologize for the confusion about the magnet units.  For this study, we used commercial permanent Magnets (Nd2Fe14B) with dimensions of 40×30×4 mm. We modified this information in the revision version of our manuscript in lines 118-119.

Comments and Suggestions for Authors. None of my objections was addressed – for accepting the manuscript, at least leaking experiments with pure (distilled) water should be done to show real “leaking” ability of the material, and not only sorption/ion exchange of ions from "real" water, as was studied.

Authors' response. We appreciate the suggestion of the reviewer. Since our objective was to evaluate the leaching of specific ions in wastewater, we did not consider pure water because we quantified the wastewater before and after passing the filter.

Other authors like Chunfeng et al., 2009 tested the potential of zeolite as an absorbent for heavy metals in wastewater, and they did not use pure water to demonstrated the adsorbent efficiency of the material, similar to our methodology. Bolan et al., 2003 evaluated the adsorption of untreated zeolites using the fellmongery effluent to measure the adsorption of ammonium ions, and they did not mention using pure water to show the real adsorption the effluent was quantified before and after the treatment, like in our study.

No additional information was included in the revised version.

Chunfeng, W., Jiansheng, L. I., Xia, S. U. N., Lianjun, W. A. N. G., & Xiuyun, S. U. N. (2009). Evaluation of zeolites synthesized from fly ash as potential adsorbents for wastewater containing heavy metals. Journal of environmental sciences, 21(1), 127-136.

Bolan, N. S., Mowatt, C., Adriano, D. C., & Blennerhassett, J. D. (2003). Removal of ammonium ions from fellmongery effluent by zeolite. Communications in soil science and plant analysis, 34(13-14), 1861-1872.

Comments and Suggestions for Authors. To my final objection to the previous version – there was a misunderstanding between me and the authors. I recommended to decrease a number of valid digits, as 5 digits are too much and it is behind accuracy of analysis, as accuracy is usually 3 digits (not decimal places). I.e., use e.g. 276 ppm, 27.6 ppm, 2.76 ppm, and not 276.00 and 27.60...

Authors' response. We the authors agree, therefore, all data had been change utilized two digits.

Comments and Suggestions for Authors. For service to reader, starting level (concentration) of ions should be given in charts, too.

Authors' response. The authors did not consider that including the initial level (concentration) of ions in the charts will help understand the results better. We want to point out that the initial levels of each treatment in each filtration time are included in the manuscript.

Warm regards.

Jesus Ochoa-Rivero
